# Patterns of symptoms possibly indicative of cancer and associated help-seeking behaviour in a large sample of United Kingdom residents—The USEFUL study

**Philip C. Hannaford**[1]*, **Alison J. Thornton**[1], **Peter Murchie**[1], **Katriina L. Whitaker**[2], **Rosalind Adam**[1], **Alison M. Elliott**[3]

1 Centre for Academic Primary Care, Institute of Applied Health Sciences, University of Aberdeen, Aberdeen, United Kingdom, 2 School of Health Sciences, University of Surrey, Guildford, Surrey, United Kingdom, 3 Graduate School, Abertay University, Dundee, United Kingdom

* p.hannaford@abdn.ac.uk

**Data Availability Statement:** The study did not receive ethics approval, or participant consent, to

## Abstract

### Background

Cancer awareness campaigns aim to increase awareness of the potential seriousness of signs and symptoms of cancer, and encourage their timely presentation to healthcare services. Enhanced understanding of the prevalence of symptoms possibly indicative of cancer in different population subgroups, and associated general practitioner (GP) help-seeking behaviour, will help to target cancer awareness campaigns more effectively.

### Aim

To determine: i) the prevalence of 21 symptoms possibly indicative of breast, colorectal, lung or upper gastrointestinal cancer in the United Kingdom (UK), including six 'red flag' symptoms; ii) whether the prevalence varies among population subgroups; iii) the proportion of symptoms self-reported as presented to GPs; iv) whether GP help-seeking behaviour varies within population subgroups.

### Methods

Self-completed questionnaire about experience of, and response to, 25 symptoms (including 21 possibly indicative of the four cancers of interest) in the previous month and year; sent to 50,000 adults aged 50 years or more and registered with 21 general practices in Staffordshire, England or across Scotland.

### Results

Completed questionnaires were received from 16,778 respondents (corrected response rate 34.2%). Almost half (45.8%) of respondents had experienced at least one symptom possibly indicative of cancer in the last month, and 58.5% in the last year. The prevalence of individual symptoms varied widely (e.g. in the last year between near zero% (*vomiting up*

place a study dataset in the public domain. The data used in this study can be made accessible to qualified researchers upon reasonable request pursuant to any restrictions required to ensure the privacy of human subjects involved. Access to data will be subject to a data sharing agreement approved by University of Aberdeen. Researchers interested in accessing USEFUL data should send their request to the study's PI, Professor Philip Hannaford (p.hannaford@abdn.ac.uk).

**Funding:** The study was funded by a project grant to Dr Alison M Elliott, Professor Philip C Hannaford, Professor Peter Murchie and Dr Katriina L Whitaker (C45810/A17927) from Cancer Research UK. The funders had no role in study design, data collection and analysis, decision to publish, or preparation of the manuscript.

**Competing interests:** The authors have declared that no competing interests exist.

blood) and 15.0% (*tired all the time*). Red flag symptoms were uncommon. Female gender, inability to work because of illness, smoking, a history of a specified medical diagnosis, low social support and lower household income were consistently associated with experiencing at least one symptom possibly indicative of cancer in both the last month and year. The proportion of people who had contacted their GP about a symptom experienced in the last month varied between 8.1% (*persistent cough*) and 39.9% (*unexplained weight loss*); in the last year between 32.8% (*hoarseness*) and 85.4% (*lump in breast*). Nearly half of respondents experiencing at least one red flag symptom in the last year did not contact their GP about it. Females, those aged 80+ years, those unable to work because of illness, ex-smokers and those previously diagnosed with a specified condition were more likely to report a symptom possibly indicative of cancer to their GP; and those on high household income less likely.

## Conclusion

Symptoms possibly indicative of cancer are common among adults aged 50+ years in the UK, although they are not evenly distributed. Help-seeking responses to different symptoms also vary. Our results suggest important opportunities to provide more nuanced messaging and targeting of symptom-based cancer awareness campaigns.

## Introduction

Although there has been important progress in recent years, the United Kingdom (UK) continues to have poorer cancer survival rates than similarly developed countries in Europe and beyond [1]. Current national cancer strategies in England [2] and Scotland [3] include cancer awareness campaigns as part of a range of actions aimed at closing this survival gap. Cancer awareness campaigns seek to increase the public's awareness of the link between certain signs and symptoms and cancer, and encourage their prompt presentation to healthcare services. The need for such initiatives is indicated partly by evidence that many people in the UK rarely consider their symptoms to be a possible indicator of cancer [4], even in high risk groups such as smokers [5–8]. Indeed there may be a general lack of awareness of cancer symptoms among the UK population [9–12].

Effective targeting of cancer awareness campaigns requires contemporaneous information about the prevalence and distribution of cancer-related symptoms within the general population, and associated patterns of help-seeking behaviour- in the UK particularly general practitioner (GP) contact. Studies from Denmark show that cancer-related symptoms are common in the general population. A cross-sectional study conducted in 2007 of 13,777 adults older than 19 years and living in Funen, Denmark, reported that 15% of participants had at least one of four common cancer red flag symptoms (also known as alarm symptoms) in the preceding 12 months [13]. A larger, predominantly internet-based survey undertaken in 2012 of 49,706 adults older than 19 years and residing across Denmark, found that 90% experienced in the preceding four weeks at least one of 44 symptoms, including red flag symptoms of lung, gastro-intestinal, gynaecological and urogenital cancer; mean number of any symptom 5.4 (Standard Deviation, SD not given) [14].

In the UK, a pooled analysis of two primary-care based surveys involving 3,756 adults older than 49 years found that 46% of respondents had experienced at least one of 10 cancer red flag

symptoms in the previous three months; mean number of symptoms 1.73 (SD 1.17) [15]. Overall, a third of people with symptoms had not contacted their GP about them, with the proportion seeking help varying by symptom experienced and characteristics of respondents. These findings suggest important opportunities for the targeting of cancer awareness campaigns. Key limitations of these UK studies, however, include the use of symptoms from the Cancer Awareness Measure [16] rather than those specific to cancer site (e.g. rectal bleeding for colorectal cancer or lump in breast for breast cancer), the small sample size and the relatively small number of respondent characteristics assessed. Another UK general population study of the prevalence of 25 physical and psychological symptoms in working-age adults provided limited information about cancer-related symptoms [17, 18].

The Understanding Symptom Experiences Fully (USEFUL) study aims to improve understanding of the prevalence, patterning and response to symptoms associated with breast, colorectal, lung and upper gastrointestinal cancer in older adults living in the UK general population. We report here its findings in relation to the experienced prevalence of symptoms possibly indicative of these four cancers, the characteristics of people experiencing those symptoms, the levels of self-reported GP help-seeking and the characteristics of those taking this action.

## Methods

### Ethics statement

National Research Ethics Service Committee East Midlands- Derby (REC reference 14/EM/1124. IRAS Project ID 160441) confirmed its favourable ethical opinion of the study, and each relevant National Health Service authority gave Research and Development management approval before the survey began. All participants received written information about the study; participants were deemed to have been given consent to participate by returning a completed questionnaire.

### Study design

The USEFUL study was underpinned by theoretical models developed to understand response to symptoms. The Model of Pathways to Treatment [19] provided a framework for distinguishing key events in the cancer diagnostic pathway. A framework integrating understanding from three process models of response to symptoms, the Commonsense Self-Regulation Model (CSM) [20], the Illness Action Model [21] and the Network Episode Model [22] guided investigation of the way in which symptoms and responses to them are interpreted and evaluated by people experiencing them.

The first phase of the USEFUL study involved surveying a large community-based sample of adults aged 50 years or more. This age group was chosen because of its higher risk of cancer and its frequent focus for cancer prevention interventions such as screening programmes or cancer awareness campaigns.

### Questionnaire development

A questionnaire for self-completion was developed to explore symptoms possibly indicative of breast, colorectal, lung and upper gastrointestinal cancer. Breast, colorectal and lung cancer were chosen as these are amongst the four most common cancers in the UK; all usually present symptomatically. Upper gastrointestinal cancer was included because it is a relatively common cancer often associated with non-specific symptoms and long diagnostic intervals. All of the chosen cancers have been the subject of cancer awareness campaigns conducted in the UK.

The choice of symptoms for inclusion in the questionnaire was guided by a review of academic literature and current clinical guidelines, and discussion with academic and clinical colleagues. The wording of each selected symptom was guided by academic literature, academic clinicians and, crucially, lay members of the University of Aberdeen College of Life Sciences and Medicine Patient Engagement Group, who provided key input into questionnaire format and wording, particularly whether the symptom descriptors captured lay understanding of each symptom. Five of the chosen symptoms included the word 'persistent' in an attempt to differentiate between symptoms that clinicians or campaigns may consider more serious and more likely to be indicative of cancer rather than a self-limiting illness. The final questionnaire included two symptoms possibly indicative of breast cancer, four of colorectal cancer, seven of lung cancer and five of upper gastrointestinal cancer (Table 1). Six symptoms (*difficulty swallowing*, *unexplained weight loss*, *coughing up blood*, *blood in stool or rectal bleeding*, *vomiting up blood*, *lump in breast* were red flag symptoms (alarm or warning symptoms and/or signs that suggest a potentially serious underlying disease.) These red flag symptoms were based on those highlighted in cancer referral guidelines. The questionnaire also asked about three 'non-specific' symptoms possibly indicative of cancer at any site; and four more general, 'masking', symptoms less likely to be indicative of cancer (to help conceal the focus of the questionnaire). In a further attempt to avoid biasing responses we broke up the ordering of symptoms. The questionnaire was piloted in two general practices in England and Scotland, and subsequently modified slightly before the main mailing. S1 File details the full content of the questionnaire.

**Table 1. Symptom description and type included in the questionnaire, listed as presented in the questionnaire.**

| Headaches | Masking symptom |
| --- | --- |
| Persistent indigestion/heartburn | Upper gastrointestinal cancer associated |
| Difficulty swallowing !! | Upper gastrointestinal cancer associated |
| Stomach or abdominal pain | Upper gastrointestinal cancer associated |
| Chest pain | Lung cancer associated |
| Hoarseness | Lung cancer associated |
| Loss of appetite | Non-specific cancer associated |
| Unexplained weight loss !! | Non-specific cancer associated |
| Persistent cough | Lung cancer associated |
| Change in ongoing cough | Lung cancer associated |
| Persistent diarrhoea | Colorectal cancer associated |
| Persistent constipation | Colorectal cancer associated |
| Coughing up phlegm | Lung cancer associated |
| Coughing up blood !! | Lung cancer associated |
| Shortness of breath | Lung cancer associated |
| Wheezy chest | Masking symptom |
| Change in bladder habits | Masking symptom |
| Change in bowel habits | Colorectal cancer associated |
| Blood in stool or rectal bleeding !! | Colorectal cancer associated |
| Back or joint pain | Masking symptom |
| Persistent vomiting | Upper gastrointestinal cancer associated |
| Vomiting up blood !! | Upper gastrointestinal cancer associated |
| Lump in breast !! | Breast cancer associated |
| Breast change other than lump | Breast cancer associated |
| Tired all the time | Non-specific cancer associated |

!! = red flag cancer symptom

## Data collection

Between May 2015 and January 2016, invitation letters were sent on behalf of 21 general practices (10 in Staffordshire, England and 11 across Scotland), to 50,000 adults aged 50 years or more. The sample size was based on a desire for reasonable precision for the annual prevalence of common symptoms (e.g. *a priori* point estimate for *lump in breast* of 3.3%, 99% confidence interval 3.0 to 3.6, assuming a response rate of 40%).

Practices were identified by two primary care research networks asked to recruit practices with different levels of rurality and deprivation. Coordinators from the networks ran searches to identify eligible patients and applied pre-specified exclusion criteria (dementia, learning disabilities, living in a nursing home or receiving palliative care). GPs in each practice were then asked to check the lists for any other patients they wished to exclude. The reason for, and number of, such exclusions was not collected. In practices with fewer than 2,500 patients aged 50 + years, all patients were identified and screened. In larger practices, a random sample of those aged 50+ years was taken to achieve the overall required sample size.

Eligible individuals remaining after exclusions were sent a study information leaflet and invitation to complete an online questionnaire. The invitees were told that the study's purpose was to enquire about symptoms experienced, their effects, participants' thoughts about the symptoms, and actions taken. Cancer was not mentioned, to avoid anxiety and minimise the risk of biasing responses. Non-respondents received reminders three and six weeks after the initial invitation letter. Reminders included a printed questionnaire to enable completion by post if preferred. Patients were able to opt-out by a study telephone number, e-mail or return of a blank questionnaire.

## Data management

In order to maintain invitee confidentiality, study packs were dispatched by the University of Aberdeen Data Management Team in Scotland and by the West Midlands Clinical Research Network in England. Questionnaires (identified by participant ID only) were returned to the research team at the Centre for Academic Primary Care, University of Aberdeen. The Data Management Team had access to the full postcode of respondents, to assign rurality and deprivation score. Postcode data were removed from the main dataset before it was forwarded to the research team for analysis. For participants from England we used the English higher level geographies Rural-Urban Classification 2011 [23] and Index of Multiple Deprivation (IMD) for England 2015 [24], and for those from Scotland, the Scottish Government 6-fold Urban Rural Classification 2013/2014 [25] and Scottish Index of Multiple Deprivation (SIMD) 2012 [26].

## Data analysis

After data cleaning, differences between groups with categorical data was assessed using the Chi-squared test. The proportion of individuals overall and in different subgroups reporting individual or combinations of symptoms possibly indicative of different cancers was calculated using SPSS Statistics version 24 [27], and Allto consulting software [28] for surrounding 99% confidence intervals. The chance of reporting symptoms possibly indicative of cancer, and associated GP help-seeking, in different population subgroups was estimated using binary logistic regression in SPSS, to calculate unadjusted and adjusted odds ratios, and their surrounding 99% confidence intervals. We estimated 99% confidence intervals because of the large number of proportions and odds ratios calculated. Variables chosen *a priori* for inclusion in the adjusted model were based on previous research [17, 18]: gender, age, marital status, social support, education, employment, household income, smoking, rurality and history of

any of the medical conditions specified in the questionnaire. All of the variables were entered simultaneously into the model To enable the entire dataset to be used in the adjustments, the six Scottish levels and three English levels of each country's rural-urban classification were combined into a single nine-level categorical variable, with Scottish large urban areas as the referent group. We used the area-based deprivation variables to examine whether response rates were related to this characteristic. We did not, however, use them in the adjusted models as other, individual-based, measures of socioeconomic status were available for respondents. It should be noted that the denominator for proportions sometimes changed because of missing values or inconsistent responses that could not be reconciled. In the analyses looking at GP help-seeking, symptomatic people with information missing about contact with their GP were assumed not to have seen their family doctor, and so were combined with those responding no to questions about this action.

## Results

### Response

After three mailings, 16,778 completed questionnaires were returned (corrected response rate after removing 403 deaths or de-registrations and 565 undelivered questionnaires 34.2%; range between practices 18.1% to 45.7%). Roughly a third of all questionnaires were completed on-line (5182, 30.9%), with a higher proportion from respondents in Scotland than England (35.1% vs 26.2%; $x^2$ = 155.4, p<0.001). In both Scotland and England significantly higher response rates were achieved from women, those living in more rural areas and those in less deprived areas (S1 Table). The response rate in England increased significantly with age from 50–54 to 70–74 years before declining in older age groups; there were no significant age differences in Scotland.

### Characteristics of respondents

More than half of all respondents were female (53.8%), younger than 70 years (71.0%), married or living with a partner (74.4%), had high social support (58.5%), had a professional, degree or postgraduate qualification (50.6%), were retired (53.4%), had never smoked (54.0%) and lived in an urban area (England 70.8%, Scotland 52.7%), Table 2. Most participants indicated a history of having been diagnosed with at least one of the conditions specified in the questionnaire (78.9%). The most common conditions were high blood pressure (32.8% of all respondents), arthritis/rheumatic disorder (23.5%) and stomach/digestive disorders (18.4%). Roughly a tenth of respondents (10.6%) reported having ever been diagnosed with cancer. Nearly all respondents reported being in good (35.0%), very good (36.1%) or excellent (11.4%) health.

Most (76.0%) respondents said that we could review their medical records, 39.4% agreed to a telephone interview and 66.1% agreed to being contacted about future studies.

### Total number of symptoms experienced

Over two thirds (69.4%) of all respondents had experienced at least one of the 25 symptoms in the last month; 26.5% had 3 or more symptoms (Table 3). These figures were almost identical when participants reporting a history of cancer were excluded (69.1% and 26.1% respectively). Since this was the case, and since people diagnosed with one cancer would still be targets for cancer awareness campaigns for other cancers, we have presented results based on replies from all respondents. Nearly half (45.8%) of all respondents had at least one symptom possibly indicative of the four cancers in the last month; 13.3% had 3 or more such symptoms, mean

**Table 2. Selected socio-demographic characteristics of respondents in the study.**

| Demographic group | Sub-group | N | % |
|---|---|---|---|
| **Sex** | Male | 7745 | 46.2 |
| | Female | 9033 | 53.8 |
| **Age group** | 50–59 | 5564 | 33.1 |
| | 60–69 | 6350 | 37.9 |
| | 70–79 | 3712 | 22.1 |
| | 80+ | 1152 | 6.9 |
| **Marital status** | Single | 1114 | 6.8 |
| | Married/living together | 12272 | 74.4 |
| | No longer married | 3100 | 18.8 |
| **Social support** | Low | 910 | 5.8 |
| | Medium | 5572 | 35.4 |
| | High | 9263 | 58.8 |
| **Educational status** | No educational qualifications | 2050 | 12.6 |
| | Secondary school or equivalent | 5356 | 32.8 |
| | College/vocational courses and other | 660 | 4.0 |
| | Professional qualification | 4250 | 26.1 |
| | Degree or postgraduate qualification | 3996 | 24.5 |
| **Employment status** | Working full-time | 3870 | 23.5 |
| | Working part-time | 1598 | 9.7 |
| | Self-employed | 1226 | 7.5 |
| | Retired | 8780 | 53.4 |
| | Unable to work due to illness/disability | 423 | 2.6 |
| | Others not in paid employment | 552 | 3.4 |
| **Household income** | < £15,000 | 3561 | 24.0 |
| | £15,000–29,999 | 4523 | 30.5 |
| | £30,000–49,999 | 3650 | 24.6 |
| | >£50,0000 | 3106 | 20.9 |
| **Smoking status** | Never smoked | 8872 | 54.0 |
| | Ex-smoker | 6095 | 37.1 |
| | Current smoker | 1471 | 8.9 |
| **Urban Rural Classification Scotland** | Large urban areas | 3234 | 36.5 |
| | Other urban areas | 1434 | 16.2 |
| | Accessible small towns | 640 | 7.2 |
| | Remote small towns | 1569 | 17.7 |
| | Accessible rural | 959 | 10.8 |
| | Remote rural | 1013 | 11.4 |
| **Rural Urban Classification England** | Urban with city and town | 2367 | 30.1 |
| | Urban with significant rural | 3201 | 40.7 |
| | Largely rural | 2293 | 29.2 |
| **Diagnosis of specified condition †** | No | 3534 | 21.1 |
| | Yes | 13244 | 78.9 |

† ever been diagnosed with asthma, cancer, epilepsy, chronic bronchitis/COPD, other chest disorder, heart disorder, stroke, diabetes, high blood pressure, liver disorder, arthritis/rheumatic disorder, mental health disorder, thyroid disorder, stomach/digestive disorder, other condition (to be specified).

**Table 3. Number of symptoms experienced in the last month and year by the respondents.**

| | All respondents (N = 16778) | | | | Respondents without a diagnosis of cancer (n = 15001) | | | |
|---|---|---|---|---|---|---|---|---|
| | All 25 symptoms | | All 21 symptoms possibly indicative of cancer | | All 25 symptoms | | All 21 symptoms possibly indicative of cancer | |
| | n | % | n | % | n | % | n | % |
| **Symptoms in last month** | | | | | | | | |
| 0 | 5142 | 30.6 | 9095 | 54.2 | 4641 | 30.9 | 8224 | 54.8 |
| 1 | 4282 | 25.5 | 3580 | 21.3 | 3858 | 25.7 | 3199 | 21.3 |
| 2 | 2918 | 17.4 | 1860 | 11.1 | 2598 | 17.3 | 1649 | 11.0 |
| 3 | 1774 | 10.6 | 994 | 5.9 | 1587 | 10.6 | 863 | 5.8 |
| 4 | 973 | 5.8 | 544 | 3.2 | 860 | 5.7 | 468 | 3.1 |
| 5 | 635 | 3.8 | 319 | 1.9 | 548 | 3.7 | 284 | 1.9 |
| >5 | 1054 | 6.3 | 386 | 2.3 | 909 | 6.1 | 314 | 2.1 |
| Mean (SD) | 1.82 (2.09) | | 1.01 (1.57) | | 1.79 (2.06) | | 0.98 (1.53) | |
| **Symptoms in last year** | | | | | | | | |
| 0 | 3357 | 20.0 | 6968 | 41.5 | 3031 | 20.2 | 6319 | 42.1 |
| 1 | 3503 | 20.9 | 3715 | 22.1 | 3166 | 21.1 | 3334 | 22.2 |
| 2 | 3114 | 18.6 | 2365 | 14.1 | 2783 | 18.6 | 2093 | 14.0 |
| 3 | 2225 | 13.3 | 1448 | 8.6 | 1982 | 13.2 | 1284 | 8.6 |
| 4 | 1484 | 8.8 | 893 | 5.3 | 1332 | 8.9 | 780 | 5.2 |
| 5–7 | 2190 | 13.1 | 1092 | 6.5 | 1936 | 12.9 | 957 | 6.4 |
| 8–10 | 649 | 3.9 | 250 | 1.5 | 569 | 3.8 | 199 | 1.3 |
| >10 | 256 | 1.5 | 47 | 0.3 | 202 | 1.3 | 35 | 0.2 |
| Mean (SD) | 2.61 (2.55) | | 1.51 (1.96) | | 2.58 (2.51) | | 1.47 (1.92) | |

SD = Standard Deviation

number 1.01 (SD 1.57) (Table 3). Symptom prevalence was slightly higher when the time frame of last year was used; 80.0% of all respondents experienced at least one of the 25 symptoms and 58.5% at least one symptom possibly indicative of the four cancers (mean number 1.51, SD 1.96).

## Individual symptoms experienced

There was a wide range in the reported prevalence of individual symptoms possibly indicative of cancer; for example, in the last month between near zero (*vomiting up blood*) and 15.0% (*tired all the time*) (Table 4). Red flag symptoms were at the lower end of the prevalence range. Even so, 7.4% of respondents reported experiencing at least one of the six red flag symptoms in the last month, and 12.7% in the last year.

More than a third (35.0%) of all respondents reported experiencing at lfeast one symptom possibly indicative of lung cancer in the last year (Table 4). The corresponding figure for upper gastrointestinal cancer symptoms was 29.4%, non-specific cancer symptoms 21.3% and colorectal cancer symptoms 17.3%. Breast cancer symptoms were less common: 3.6% in females and 0.3% in males. Among the 5,880 participants who reported experiencing at least one symptom possibly indicative of lung cancer in the last year, 4,041 (68.7%) had not previously been diagnosed with asthma, chronic bronchitis/COPD or other chest disorder. Similarly, among the 4,929 participants who reported experiencing at least one symptom possibly indicative of upper gastrointestinal tract cancer in the last year, 3,009 (61.0%) had never been diagnosed as having a stomach/digestive disorder.

**Table 4. Proportion of respondents having experienced symptoms possibly indicative of different cancers in the last month and year, and proportion of those with the symptoms who contacted their GP about it.**

| | In last month | | In last year: | |
| --- | --- | --- | --- | --- |
| | Had symptom | Contacted GP in last month[¥] | Had symptom | Contacted GP about symptom in last year[¥] |
| | n /16778 (%, 99% CI) | n/S (%, 99% CI) | n /16778 (%, 99% CI) | n/S (%, 99% CI) |
| Persistent indigestion/heartburn | 1969 (11.7, 11.1–12.3) | 330 (16.8, 14.6–19.0) | 2769 (16.5, 15.8–17.2) | 1221 (44.1, 41.7–46.5) |
| Difficulty swallowing !! | 573 (3.4, 3.0–3.8) | 103 (18.0, 13.9–22.1) | 884 (5.3, 4.9–5.6) | 327 (37.0, 32.8–41.2) |
| Stomach or abdominal pain | 1661 (9.9, 9.3–10.5) | 427 (25.7, 22.9–28.5) | 2734 (16.3, 15.6–17.0) | 1322 (48.4, 45.9–50.9) |
| Persistent vomiting | 36 (0.2, 0.1–0.3) | 18 (50.0, 28.5–71.5) | 95 (0.6, 0.5–0.8) | 57 (60.0, 47.1–73.0) |
| Vomiting up blood !! | 3 (0.0, 0.0–0.0) | 0 (-) | 12 (0.1, 0.0–0.2) | 7 (58.3, 21.6–95.0) |
| *At least one upper GI tract cancer symptom* | *3418 (20.4, 19.6–21.2)* | *715 (20.9, 19.1–22.7)* ◊ | *4929 (29.4, 28.5–30.3)* | *2464 (50.0, 48.2–51.8)* ◊ |
| Chest pain | 908 (5.4, 5.0–5.9) | 283 (31.2, 27.2–35.2) | 1490 (8.9, 8.3–9.5) | 842 (56.5, 53.2–59.8) |
| Hoarseness | 775 (4.6, 4.2–5.0) | 117 (15.1, 11.8–18.4) | 1319 (7.9, 7.4–8.4) | 378 (28.7, 25.5–31.9) |
| Persistent cough | 1356 (8.1, 7.6–8.6) | 378 (27.9, 24.8–31.0) | 2189 (13.0, 12.3–13.7) | 1101 (50.3, 47.6–53.1) |
| Change in ongoing cough | 188 (1.1, 0.9–1.3) | 62 (33.0, 24.2–41.8) | 298 (1.8, 1.5–2.1) | 145 (48.7, 41.2–56.2) |
| Coughing up phlegm | 1481 (8.8, 8.2–9.4) | 288 (19.4, 16.8–22.1) | 2241 (13.4, 12.7–14.1) | 850 (37.9, 35.3–40.5) |
| Coughing up blood !! | 36 (0.2, 0.1–0.3) | 10 (27.8, 8.6–47.0) | 91 (0.5, 0.4–0.6) | 60 (65.9, 53.1–78.7) |
| Shortness of breath | 1989 (11.9, 11.3–12.5) | 562 (28.3, 25.7–30.9) | 2647 (15.8, 15.1–16.5) | 1419 (53.6, 51.1–56.1) |
| *At least one lung cancer symptom* | *4194 (25.0, 24.1–25.9)* | *1024 (24.4, 22.7–26.1)* ◊ | *5880 (35.0, 34.1–36.0)* | *3131 (53.2, 50.9–55.5)* ◊ |
| Persistent diarrhoea | 499 (3.0, 2.7–3.3) | 115 (23.0, 18.2–27.9) | 729 (4.3, 3.9–4.7) | 343 (47.1, 42.3–51.9) |
| Persistent constipation | 726 (4.3, 3.9–4.7) | 129 (17.8, 14.1–21.5) | 964 (5.7, 5.2–6.2) | 435 (45.1, 41.0–49.2) |
| Change in bowel habits | 837 (5.0, 4.6–5.4) | 235 (28.1, 24.1–32.1) | 1323 (7.9, 7.4–8.4) | 641 (48.5, 45.0–52.0) |
| Blood in stool or rectal bleeding !! | 487 (2.9, 2.6–3.2) | 94 (19.3, 14.7–23.9) | 872 (5.2, 4.8–5.6) | 399 (45.8, 41.5–51.2 |
| *At least one colorectal cancer symptom* | *2013 (12.0, 11.4–12.7)* | *416 (20.7, 18.4–23.0)* ◊ | *2897 (17.3, 16.6–18.1)* | *1404 (48.5, 45.1–51.9)* ◊ |
| Lump in breast !! | 77 (0.5, 0.4–0.6) | 24 (31.2, 17.6–44.8) | 175 (1.0, 0.8–1.2) | 135 (77.1, 68.9–85.3) |
| Breast change other than lump | 110 (0.7, 0.5–0.9) | 42 (38.2, 26.3–50.1) | 208 (1.2, 1.0–1.4) | 132 (63.5, 54.9–72.1) |
| *At least one breast cancer symptom* | *178 (1.1, 0.9–1.3)* ‡ | *63 (35.4, 26.2–44.6)* ◊ | *355 (2.1, 1.8–2.4)* ‡‡ | *259 (73.0, 66.9–79.1)* ◊ |
| Loss of appetite | 478 (2.8, 2.5–3.1) | 100 (20.9, 16.1–25.7) | 883 (5.3, 4.9–5.8) | 276 (31.3, 27.3–35.3) |
| Unexplained weight loss !! | 158 (0.9, 0.7–1.1) | 63 (39.9, 29.9–49.9) | 341 (2.0, 1.7–2.3) | 189 (55.4, 48.5–62.3) |
| Tired all the time | 2516 (15.0, 4.3–15.7) | 570 (22.7, 20.6–24.9) | 3078 (18.3, 17.5–19.1) | 1300 (42.2, 39.9–44.5) |
| *At least one non-specific cancer symptom* | *2756 (16.4, 15.7–17.1)* | *630 (22.9, 20.8–25.0)* ◊ | *3577 (21.3, 20.5–22.1)* | *1639 (45.8, 42.6–49.0)* ◊ |
| *At least one red flag symptom* | *1239 (7.4, 6.9–7.9)* | *278 (22.4, 19.4–25.5)* ◊ | *2131 (12.7, 12.0–13.4)* | *1085 (50.9, 48.1–53.7)* ◊ |
| *At least one cancer symptom* | *7683 (45.8, 44.8–46.8)* | *2038 (26.5, 25.2–27.8)* ◊ | *9810 (58.4, 57.4–59.4)* | *5836 (59.5, 58.5–60.5)* ◊ |

[¥] Symptomatic people with information missing about contact with their GP were assumed to have not seen their family doctor

n = number, n/S = number contacting GP/respondents experiencing this symptom, CI = confidence interval

!! = red flag cancer symptom

◊ proportion who saw GP for at least one of the symptoms in this group

‡ 159/9033 (1.8, 1.4–2.2) in females

‡‡ 328/9033 (3.6, 3.1–4.1) in females

## Characteristics of those experiencing symptoms

Compared with their respective referent group, women, those unable to work because of illness or disability, ex- and current-smokers and those with a history of a specified condition were significantly more likely to report experiencing at least one symptom possibly indicative of the four cancers of interest; in both time periods and after adjusting for other factors (Table 5). On the other hand, those aged 60 to 69 years, those with medium or high social support and those in a household within an income of at least £15,000 were significantly less likely

**Table 5. Proportion and chance of participants in different subgroups reporting having experienced at least one symptom possibly indicative of cancer in the last month and last year.**

| Sub-group† | In the last month | | | In the last year | | |
|---|---|---|---|---|---|---|
| | % (99% CI) | UOR (99% CI) | AOR* (99% CI) | % (99% CI) | UOR (99% CI) | AOR* (99% CI) |
| **Sex** (Male)[R] | 44.0 (42.6–45.5) | | | 56.3 (54.9–57.8) | | |
| Female | 47.3 (46.0–48.7) | **1.15 (1.06–1.24)** | **1.19 (1.08–1.31)** | 60.3 (59.0–61.6) | **1.18 (1.09–1.28)** | **1.19 (1.08–1.32)** |
| **Age** (50–59)[R] | 44.6 (43.9–46.3) | | | 60.1 (58.4–61.8) | | |
| 60–69 | 43.3 (41.7–44.9) | 0.95 (0.86–1.04) | **0.84(0.73–0.96)** | 56.0 (54.4–57.6) | **0.85 (0.77–0.93)** | **0.75 (0.66–0.86)** |
| 70–79 | 47.1 (45.0–49.2) | 1.10 (0.99–1.23) | 0.91 (0.77–1.09) | 57.3 (55.2–59.4) | **0.89 (0.80–1.00)** | **0.74 (0.62–0.88)** |
| 80+ | 60.9 (57.2–64.6) | **1.93 (1.63–2.89)** | **1.49 (1.16–1.91)** | 68.3 (64.8–71.8) | **1.43 (1.20–1.71)** | 1.12 (0.87–1.45) |
| **Marital status** (Single)[R] | 47.1 (56.0–64.6) | | | 59.8 (56.0–63.6) | | |
| Married/living together | 43.9 (42.8–45.1) | 0.88 (0.75–1.03) | 1.07 (0.88–1.30) | 56.9 (55.8–58.1) | 0.89 (0.75–1.05) | 1.05 (0.87–1.28) |
| No longer married | 52.9 (50.6–55.2) | **1.26 (1.05–1.51)** | 1.16 (0.94–1.43) | 64.6 (62.4–66.8) | **1.23 (1.02–1.48)** | 1.19 (0.97–1.48) |
| **Social support** (Low)[R] | 54.8 (50.6–59.1) | | | 66.4 (62.4–70.4) | | |
| Medium | 46.7 (45.0–48.4) | **0.72 (0.60–0.87)** | **0.80 (0.65–0.99)** | 58.8 (57.1–60.5) | **0.72 (0.60–0.88)** | **0.78 (0.63–0.96)** |
| High | 44.6 (43.3–45.9) | **0.66 (0.55–0.79)** | **0.75 (0.61–0.92)** | 57.9 (56.6–59.2) | **0.70 (0.58–0.84)** | **0.76 (0.61–0.94)** |
| **Education** (No qualifications)[R] | 51.9 (49.1–54.7) | | | 60.7 (57.9–63.5) | | 0.76 (0.61–0.94) |
| Secondary school or equivalent | 47.6 (45.8–49.4) | **0.84 (0.74–0.96)** | 1.00 (0.84–1.17) | 59.0 (57.3–60.7) | 0.93 (0.81–1.07) 1. | 1.06 (0.90–1.25) |
| College/vocational courses and other | 49.4 (44.4–54.4) | 0.90 (0.72–1.14) | 0.95 (0.73–1.25) | 62.1 (57.2–67.0) | 1.06 (0.84–1.35) | 1.05 (0.80–1.38) |
| Professional qualification | 44.8 (42.8–46.8) | **0.75 (0.66–0.87)** | 0.97 (0.81–1.15) | 58.8 (56.9–20.7) | 0.92 (0.80–1.06) | 1.09 (0.91–1.30) |
| Degree or postgraduate qualification | 40.9 (38.9–42.9) | **0.64 (0.56–0.77)** | 0.91 (0.76–1.09) | 56.0 (54.0–58.0) | **0.82 (0.72–0.95)** | 1.04 (0.87–1.25) |
| **Employment** (Working full-time)[R] | 42.3 (40.3–44.4) | | | 57.5 (55.5–59.6) | | |
| Working part-time | 42.4 (39.2–45.6) | 1.00 (0.86–1.17) | 0.89 (0.75–1.07) | 56.1 (52.9–59.3) | 0.95 (0.81–1.10) | 0.88 (0.74–1.05) |
| Self-employed | 43.7 (40.1–47.4) | 1.06 (0.90–1.26) | 1.05 (0.87–1.27) | 57.7 (54.1–61.3) | 1.01 (0.85–1.20) | 1.05 (0.86–1.27) |
| Retired | 46.2 (44.8–47.6) | **1.17 (1.06–1.30)** | 0.87 (0.74–1.02) | 57.7 (56.3–59.1) | 1.01 (0.91–1.11) | 0.90 (0.77–1.05) |
| Unable to work due to illness/disability | 86.5 (82.2–90.8) | **8.77 (6.02–12.77)** | **5.50 (3.57–8.47)** | 91.3 (87.8–94.8) | **7.71 (4.91–12.10)** | **4.72 (2.87–7.76)** |
| Others not in paid employment | 47.5 (42.0–53.0) | 1.23 (0.98–1.56) | 0.98 (0.75–1.29) | 61.6 (56.3–66.9) | 1.18 (0.93–1.51) | 1.00 (0.76–1.33) |
| **Household income** (< £15,000)[R] | 54.7 (52.6–56.9) | | | 64.9 (62.8–67.0) | | |
| £15,000–29,999 | 45.4 (43.5–47.3) | **0.69 (0.61–0.77)** | **0.82 (0.72–0.94)** | 57.0 (55.1–58.9) | **0.72 (0.64–0.81)** | **0.82 (0.71–0.94)** |
| £30,000–49,999 | 42.5 (40.4–44.6) | **0.61 (0.54–0.69)** | **0.80 (0.68–0.93)** | 56.8 (54.7–58.9) | **0.71 (0.63–0.81)** | **0.83 (0.71–0.97)** |
| >£50,000 | 38.3 (36.1–40.6) | **0.51 (0.45–0.58)** | **0.68 (0.57–0.81)** | 54.6 (52.3–56.9) | **0.65 (0.57–0.74)** | **0.74 (0.62–0.88)** |
| **Smoking status** (Never smoked)[R] | 41.6 (40.3–43.0) | | | 55.0 (53.6–56.4) | | |
| Ex-smoker | 50.0 (48.4–51.7) | **1.41 (1.29–1.53)** | **1.29 (1.16–1.42)** | 61.8 (60.2–63.4) | **1.32 (1.21–1.45)** | **1.23 (1.12–1.36)** |
| Current smoker | 54.2 (50.9–57.6) | **1.67 (1.44–1.93** | **1.54 (1.31–1.83)** | 65.5 (62.3–68.7) | **1.55 (1.33–1.80)** | **1.37 (1.15–1.63)** |
| **Rural Urban** (Scotland large urban)[R] | 45.9 (43.6–48.2) | | | 60.9 (58.7–63.1) | | |
| Scotland other urban areas | 45.5 (42.1–48.9) | 0.98 (0.83–1.16) | 0.99 (0.82–1.20) | 58.6 (55.3–62.0) | 0.91 (0.77–1.07) | 0.91 (0.75–1.10) |
| Scotland accessible small towns | 45.6 (40.5–50.7) | 0.99 (0.79–1.24) | 0.92 (0.71–1.19) | 57.8 (52.8–62.8) | 0.88 (0.70–1.10) | 0.84 (0.65–1.08) |
| Scotland remote small towns | 45.4 (42-2–48.6) | 0.98 (0.84–1.15) | 1.03 (0.86–1.24) | 57.8 (54.6–61.0) | 0.88 (0.75–1.03) | 0.94 (0.78–1.13) |
| Scotland accessible rural | 43.6 (39.5–47.7) | 0.91 (0.75–1.10) | 0.98 (0.79–1.22) | 55.6 (51.5–59.7) | **0.80 (0.66–0.97)** | 0.85 (0.68–1.06) |
| Scotland remote rural | 44.5 (40.5–48.5) | 0.95 (0.78–1.14) | 0.96 (0.77–1.19) | 57.4 (53.4–61.4) | 0.87 (0.72–1.04) | 0.90 (0.73–1.12) |
| England urban with city and town | 50.5 (47.9–53.2) | **1.20 (1.04–1.38)** | 1.09 (0.92–1.30) | 61.9 (59.3–64.5) | 1.04 (0.90–1.20) | 0.96 (0.81–1.14) |
| England urban with significant rural | 43.7 (41.4–46.0) | 0.92 (0.81–1.04) | 0.94 (0.81–1.09) | 56.5 (54.2–58.8) | **0.83 (0.73–0.95)** | 0.87 (0.74–1.01) |
| England largely rural | 45.9 (43.2–48.6) | 1.00 (0.87–1.15) | 1.00 (0.85–1.18) | 56.6 (53.9–59.3) | **0.83 (0.72–0.96)** | 0.85(0.72–1.00) |
| **Diagnosis of specified condition** (No)[R] | 27.0 (25.1–28.9) | | | 40.8 (38.7–42.9) | | |
| Yes | 50.8 (49.7–51.9) | **2.80 (2.51–3.12)** | **2.65 (2.35–3.00)** | 63.2 (62.1–64.3) | **2.50 (2.26–2.76)** | **2.48 (2.21–2.78)** |

CI = confidence interval, UOR = unadjusted odds ratio, AOR = adjusted odds ratio

† Number in each subgroup as per Table 1.

*adjusted odds ratio: adjusted for gender, age, marital status, social support, education, employment, household income, smoking, rurality, ever diagnosis of specified condition, except when the variable itself is being examined.

[R] = Referent group for odds ratios

to report at least one symptom possibly indicative of cancer irrespective of what time frame was used. Marital status, education and rurality were not significant variables in the adjusted model. There were generally few statistically significant adjusted differences between subgroups in the chances of experiencing symptoms possibly indicative of the different cancers in the last year (S2 Table). Where differences did emerge, the pattern of associations was broadly consistent with that of experiencing any symptom possibly indicative of cancer, i.e. higher reporting among females, those unable to work because of illness, smokers and individuals with a history of one of the specified medical conditions; and less among those aged 60 to 79 years, those with greater social support and those with higher household incomes.

## Proportion contacting the GP

Overall, 26.5% of respondents experiencing at least one symptom possibly indicative of the four cancers in the last month contacted their GP about at least one symptom, and 59.5% in the previous year (Table 4). The corresponding proportions were smaller among participants experiencing at least one red flag symptom; 22.4% and 50.9% respectively. Contact with the GP varied greatly for individual (including red flag) symptoms; between 15.1% (*hoarseness*) and 50.0% (*persistent vomiting*) for symptoms experienced in the last month and between 28.7% (*hoarseness*) and 77.1% (*lump in breast*) in the last year.

## Characteristics of those contacting the GP

Table 6 shows the association between different characteristics of the respondents and the likelihood of contacting the GP if a symptom possibly indicative of cancer was experienced in the last month or last year. In the adjusted model, compared with their respective referent group, symptomatic women, those aged 80+ years, those unable to work due to illness or disability, ex-smokers and those with a history of a specified condition were more likely to contact their GP about at least one symptom; and people with the highest level of household income less likely to do so. The pattern of associations was the same regardless of the time frame (last month or last year) considered for experiencing symptoms.

# Discussion

## Main findings

Symptoms possibly indicative of cancer were common in the adults surveyed in our study, with nearly half experiencing at least one such symptom in the last month and nearly three-fifths in the last year. Many people experienced multiple symptoms. The prevalence of individual symptoms varied widely, with red flag symptoms at the lower end of the range. The prevalence of symptoms varied within different population subgroups. More than three quarters of respondents experiencing at least one red flag symptom in the last month did not reported it to their GP, and nearly half of those experiencing such symptoms in the last year. There were important variations in the level of GP help-seeking by symptom type and the characteristics of person experiencing them.

## Strengths and limitations

The USEFUL study is the largest investigation so far in the UK of the prevalence and GP help-seeking responses to symptoms possibly indicative of cancer in a community setting. Its large size meant that we had good precision for common symptoms. The questionnaire underwent detailed refinement during both the development phase when experts and lay members reviewed early drafts and the pilot phase which resulted in some changes to make the

**Table 6. Proportion and chance of participants in different subgroups experiencing at least one symptom possibly indicative of the four cancers and seeing their GP for at least one such symptom in the last month and last year.**

| Sub-group† | In the last month | | | In the last year | | |
|---|---|---|---|---|---|---|
| | % (99% CI) | UOR (99% CI) | AOR* (99% CI) | % (99% CI) | UOR (99% CI) | AOR* (99% CI) |
| Sex (Male)[R] | 25.2 (23.9–26.5) | | | 56.2 (54.8–57.7) | | |
| Female | 27.2 (26.0–28.4) | **1.21 (1.07–1.36)** | **1.17 (1.01–1.37)** | 62.2 (60.9–63.5) | **1.30 (1.19–1.41)** | **1.29 (1.17–1.43)** |
| Age (50–59)[R] | 24.9 (23.4–26.4) | | | 55.4 (53.7–57.1) | | |
| 60–69 | 26.2 (24.8–27.6) | 1.03 (0.88–1.12) | 0.91 (0.74–1.12) | 59.0 (57.4–60.6) | 0.99 (0.89–1.09) | 0.90 (0.78–1.03) |
| 70–79 | 27.6 (25.7–29.5) | **1.20 (1.01–1.41)** | 1.12 (0.78–1.32) | 63.8 (61.8–65.8) | **1.15 (1.03–1.29)** | 1.00 (0.84–1.20 |
| 80+ | 31.0 (27.534.5)- | **1.86 (1.49–2.32)** | **1.54 (1.10–2.16)** | 67.0 (63.4–70.6) | **1.69 (1.42–2.00)** | **1.48 (1.15–1.89)** |
| **Marital status** (Single)[R] | 26.1 (22.7–29.5 | | | 57.1 (53.3–60.9) | | |
| Married/living together | 25.3 (24.3–26.3) | 0.89 (0.70–1.14) | 1.08 (0.81–1.44) | 58.8 (57.7–59.9) | 0.97 (0.82–1.15) | 1.06 (0.87–1.29) |
| No longer married | 30.5 (28.4–32.6) | **1.38 (1.05–1.79)** | 1.24 (0.92–1.69) | 62.9 (60.7–65.1) | **1.32 (1.10–1.60)** | 1.15 (0.92–1.42) |
| **Social support** (Low)[R] | 23.2 (19.6–26.8) | | | 57.8 (53.6–62.0) | | |
| Medium | 25.7 (24.2–27.2) | 0.93 (0.71–1.23) | 1.08 (0.78–1.48) | 58.3 (56.6–60.0) | 0.84 (0.69–1.02) | 0.90 (0.73–1.12) |
| High | 27.3 (26.1–28.5) | 0.95 (0.73–1.24) | 1.12 (0.82–1.53) | 59.8 (58.6–61.2) | 0.85 (0.71–1.03) | 0.93 (0.75–1.15) |
| **Education** (No qualifications)[R] | 28.3 (25.7–30.9) | | | 64.5 (61.8–67.2) | | |
| Secondary school or equivalent | 28.1 (26.5–29.7) | 0.90 (0.74–1.09) 1.1 | 0.97 (0.77–1.22 | 62.0 (60.3–63.7) | 0.90 (0.78–1.03) | 1.02 (0.86–1.20) |
| College/vocational courses and other | 32.2 (27.5–36.9) | 1.10 (0.80–1.51) | 1.13 (0.78–1.63) | 61.2 (56.3–66.1) | 0.96 (0.75–1.21) | 1.00 (0.75–1.31) |
| Professional qualification | 25.8 (24.0–27.5) | **0.76 (0.62–0.93)** | 0.94 (0.73–1.20) | 58.3 (56.4–60.3) | **0.81 (0.70–0.94)** | 0.97 (0.81–1.16) |
| Degree or postgraduate qualification | 22.4 (20.7–24.1) | **0.58 (0.47–0.72)** | 0.82 (0.63–1.07) | 54.4 (52.4–56.4) | **0.68 (0.59–0.79)** | 0.88 (0.73–1.07) |
| **Employment** (Working full-time)[R] | 22.5 (20.8–24.2) | | | 53.5 (51.4–55.6) | | |
| Working part-time | 27.0 (24.1–29.9) | 1.23 (0.96–1.58) | 1.04 (0.79–1.38) | 58.5 (55.3–61.7) | 1.10 (0.94–1.30) | 0.98 (0.81–1.18) |
| Self-employed | 22.2 (19.1–25.3) | 1.02 (0.77–1.36) | 0.91 (0.66–1.25) | 54.1 (50.4–57.8) | 1.02 (0.85–1.23) | 0.98 (0.80–1.21) |
| Retired | 27.0 (25.8–28.2) | **1.36 (1.15–1.60)** | 0.94 (0.73–1.20) | 61.4 (60.1–62.7) | **1.24 (1.11–1.38)** | 0.93 (0.79–1.10) |
| Unable to work due to illness/disability | 43.4 (37.2–49.6) | **5.73 (4.27–7.70)** | **3.50 (2.47–4.95)** | 78.8 (73.7–83.9) | **5.75 (4.29–7.71)** | **3.92 (2.81–5.49)** |
| Others not in paid employment | 28.2 (23.3–33.1) | **1.47 (1.04–2.09)** | 1.27 (0.86–1.90) | 63.5 (58.2–68.8) | **1.45 (1.14–1.84)** | 1.24 (0.93–1.64) |
| **Household income** (< £15,000)[R] | 31.0 (29.0–33.0) | | | 64.5 (62.4–66.6) | | |
| £15,000–29,999 | 26.5 (24.8–28.2) | **0.67 (0.57–0.79)** | 0.87 (0.71–1.05) | 60.5 (58.6–62.4) | **0.73 (0.65–0.83)** | 0.91 (0.79–1.04) |
| £30,000–49,999 | 24.0 (22.2–25.8) | **0.55 (0.46–0.67)** | 0.79 (0.63–1.00) | 57.8 (55.7–59.9) | **0.68 (0.60–0.77)** | 0.93 (0.79–1.09) |
| >£50,000 | 19.9 (18.1–21.8) | **0.40 (0.33–0.50)** | **0.62 (0.47–0.82)** | 52.0 (49.7–54.3) | **0.55 (0.48–0.63)** | **0.79 (0.66–0.95)** |
| **Smoking status** (Never smoked)[R] | 25.4 (24.2–26.6) | | | 58.5 (57.2–59.9) | | |
| Ex-smoker | 27.9 (26.4–29.4) | **1.37 (1.21–1.56)** | **1.23 (1.06–1.43)** | 60.9 (50.3–62.5) | **1.27 (1.16–1.39)** | **1.16 (1.05–1.29)** |
| Current smoker | 26.6 23.6–29.6) | **1.43 (1.16–1.76)** | 1.21 (0.95–1.54) | 57.4 (54.1–60.7) | **1.27 (1.09–1.47)** | 1.13 (0.95–1.35) |
| **Rural Urban** (Scotland large urban)[R] | 28.3 (26.3–30.3) | | | 58.2 (56.0–60.4) | | |
| Scotland other urban areas | 23.6 (20.7–26.5) | 0.81 (0.62–1.04) | 0.82 (0.61–1.10) | 61.0 (57.7–64.3) | 1.01 (0.85–1.20) | 0.98 (0.81–1.20) |
| Scotland accessible small towns | 30.5 (25.8–35.2) | 1.08 (0.78–1.50) | 0.86 (0.58–1.26) | 63.0 (58.1–67.9) | 1.04 (0.83–1.31) | 0.92 (0.71–1.20) |
| Scotland remote small towns | 25.7 (22.9–28.5) | 0.89 (0.69–1.13) | 0.86 (0.65–1.13) | 58.9 (55.7–62.1) | 0.94 (0.79–1.11) | 0.92 (0.76–1.12) |
| Scotland accessible rural | 26.6 (22.9–30.3) | 0.88 (0.65–1.18) | 0.92 (0.66–1.28) | 58.3 (54.2–62.4) | 0.87 (0.71–1.07) | 0.89 (0.71–1.13) |
| Scotland remote rural | 28.8 (25.1–32.5) | 0.99 (0.75–1.30) | 1.01 (0.74–1.39) | 58.1 (54.1–62.1) | 0.91 (0.75–1.11) | 0.92 (0.73–1.15) |
| England urban with city and town | 28.5 (26.1–30.9) | 1.12 (0.92–1.38) | 0.95 (0.75–1.21) | 61.7 (59.1–64.3) | 1.13 (0.97–1.30) | 1.00 (0.84–1.19) |
| England urban with significant rural | 24.6 (22.6–26.6) | 0.81 (0.66–0.99) | 0.79 (0.63–1.00) | 59.7 (57.5–61.9) | 0.92 (0.81–1.06) | 0.89 (0.76–1.04) |
| England largely rural | 24.7 (22.4–27.0) | 0.86 (0.69–1.06) | 0.82 (0.64–1.05) | 58.2 (55.6–60.9) | 0.89 (0.77–1.04) | 0.85 (0.72–1.01) |
| **Diagnosis of specified condition** (No)[R] | 17.0 (15.4–18.6) | | | 43.0 (40.9–45.2) | | |
| Yes | 27.9 (26.9–28.9) | **3.44 (2.77–4.27)** | **2.97 (2.32–3.79)** | 62.3 (61.2–63.4) | **3.05 (2.70–3.45)** | **2.80 (2.44–3.21)** |

CI = confidence interval, UOR = unadjusted odds ratio, AOR = adjusted odds ratio

† Number in each subgroup as per Table 1.

*adjusted odds ratio: adjusted for gender, age, marital status, social support, education, employment, household income, smoking, rurality, ever diagnosis of specified condition, except when the variable itself is being examined.

[R] = Referent group for odds ratios

questionnaire easier to follow. A key consideration when discussing early drafts was whether our symptom descriptors accorded with lay understanding. The symptom components of the questionnaire were based on a questionnaire used in a previous study, which had been found to collect appropriate data [17, 18]. Questions about the specific concepts of illness perceptions and cancer awareness were taken from validated questionnaires (Brief Illness Perception Questionnaire [29] and Cancer Awareness Measure [16]). We looked at symptom experience over the past month (the main focus) and past year. The patterns of associations were the same regardless of the time frame used. It was useful to include both timeframes to ensure our findings were comparable to previous research in other countries [13].

Masking symptoms were included in the questionnaire, and the explicit mention of cancer avoided, to limit potential anxiety and minimise the biasing of responses. Even so, some recipients may have become aware of our interest in cancer, having seen a description of the study on a publically accessible website, such as that of Cancer Research UK which stated that the study was looking at cancer. Three recipients of the invitation to participate contacted us to clarify the study's purpose. Although some participants reported a history of cancer when completing the questionnaire. We have no information about cancer screening within the study sample. We found that the inclusion of people with a history of cancer did not materially affect our prevalence results. Some of the reported symptoms are likely to have arisen from other chronic conditions. Nevertheless, most cancer awareness campaigns are targeted at the total population regardless of previous or current medical history, so understanding the prevalence and pattern of cancer-related symptoms, and associated help-seeking behaviour, in the total population is essential. Previous studies investigating GP help-seeking have often focused on individuals with cancer, retrospectively examining their symptom experiences prior to diagnosis [5–7, 30–34]. Such studies are limited by variable patient recall [35] and possible changes in symptom interpretation post diagnosis [36]. Our study participants were of an age (50+ years) when the incidence of cancer rises sharply, and when many cancer initiatives, such as screening, cancer awareness campaigns or clinical guidance of prompt investigation for particular symptoms begins [37, 38]. Thus, our results may match reality more closely than those of studies involving a different age group or asking about anticipated delays for hypothetical scenarios [12, 39, 40].

We were able to consider a large number of factors in the analyses that may be related to the reporting of symptoms possibly indicative of cancer. Some consistent associations were observed, both overall and for symptoms associated with different types of cancer. The cross-sectional nature of the study, however, means that we are unable to determine whether the observed associations were causal. Furthermore, as with any observational study, residual confounding from factors (such as lifestyle) not included in the model may have occurred. This said, our observations should be informative for the targeting of future cancer symptom awareness campaigns and interventions throughout the UK.

We did not have information about how many practices approached by the research networks declined to participate, or about the characteristics of participating practices compared to non-participating practices. In addition, we have no information about the number of, and reasons for, exclusions made by the participating practices when screening the lists of potential invitees. We do not know therefore how generalizable our results are to the wider UK population.

The sampling of patients registered with a GP is a method routinely used in the UK for sampling the general population since most people are registered with a GP so that they can obtain healthcare. Non-registered individuals who were excluded from our study may have a different experience of symptoms possibly indicative of cancer than participants in our study, and probably different patterns of response to these symptoms. A key consideration when

interpreting our findings is the low response rate; a common problem with recent epidemiological research [4, 15, 17, 18, 41, 42]. We tried to mitigate against a low response rate by using a number of recommended approaches [43], including inviting participation using personalised letters on general practice-headed notepaper, offering both on-line and paper questionnaires and sending two reminders. We did not have ethics approval or resources to increase participation further, such as by using a telemarketing company to approach non-responders or holding a lottery for respondents [14]. The primary concern with low response rates is its potential to introduce bias. There was evidence of differential response by gender, age, rurality and deprivation. It is difficult to assess the overall impact of this on our findings, although there may have been some underestimation of symptom prevalence since those on lower incomes (less likely to respond) appeared more likely to report symptoms possible indicative of cancer.

As with all other investigations of symptoms, our study was based on self-reported information. Deliberately, we did not define or explain symptom descriptors such as 'persistent' or 'unexplained'. It is likely that these descriptors have different meanings for different people. In a separate exercise we have conducted qualitative interviews with a number of participants to examine their perceptions of the term 'persistence'. These findings will be published in a separate paper. Importantly, our study shows how many people perceived themselves to have such symptoms, and how they responded. We have found that participant self-report of help-seeking for symptoms possibly indicative of cancer is reasonably accurate [44]. Individuals may have differed in their understanding of what was meant by the symptoms listed, although our development work involved members of the general public in an attempt to mitigate against this. Furthermore, many people had a previous diagnosis of a number of medical conditions, reflecting the age profile of our sample. We do not know how many people had active symptomatic disease, particularly chronic disease, which may affect awareness and response to different symptoms. Where we were able to assess this (i.e. for symptoms possibly indicative of lung or upper gastrointestinal cancer) we found that most symptoms occurred in participants without a previous relevant diagnosis; suggesting that our findings were not simply due to a high proportion of participants with historic diagnoses which had been symptomatic for some time. Although subjective, symptoms are powerful drivers of healthcare service use, so understanding how people experience and react to them remains crucial.

Space constraints prevent us from presenting in-depth patterns of, and different influences on, help-seeking responses to experienced symptoms. Subsequent papers will provide this information. Nevertheless, the top-level results presented here show important variations within the general population in GP help-seeking behaviour, highlighting the need for research in this area.

## Other studies

Few studies have assessed a range of symptoms possibly indicative of cancer from a community perspective. Furthermore, comparison between studies can be difficult because of differences in the range of symptoms assessed, descriptors used and the age of participants. For example, only six of the 44 symptoms included in the Danish Symptom Cohort study [14] used the same wording as our study; with another four using broadly similar wording (Table 7). The proportion of people in the Danish study experiencing these symptoms in the preceding month was higher (albeit sometimes only marginally) for eight of the ten symptoms, than in our study. Some of the differences may be attributable to the inclusion of younger participants in the Danish study; we found in a previous UK study [18] a higher prevalence of symptoms among younger people. The Danish study also found that more people contacted

**Table 7. Comparison of the proportion of people reporting different symptoms and proportion seeking GP help in the last month, between our study and the Danish Symptom Cohort [14].**

| | Our study | | Danish Study | |
|---|---|---|---|---|
| | Had symptom | Contacted GP | Had symptom | Contacted GP |
| Same wording: | % | % | % | % |
| Difficulty swallowing | 3.4 | 18.0 | 3.5 | 34.9 |
| Hoarseness | 4.6 | 15.1 | 7.7 | 18.7 |
| Coughing up blood | 0.2 | 27.8 | 0.1 | 47.5 |
| Shortness of breath | 11.9 | 28.3 | 8.0 | 49.7 |
| Blood in stool or rectal bleeding | 2.9 | 19.3 | 4.6 | 33.7 |
| Loss of appetite | 2.8 | 20.9 | 6.3 | 19.4 |
| Slightly different wording- our study / Danish study wording: | | | | |
| Stomach or abdominal pain / abdominal pain | 9.9 | 25.7 | 19.6 | 27.8 |
| Persistent vomiting / repeated vomiting | 0.2 | 50.0 | 1.3 | 33.6 |
| Vomiting up blood / blood in vomit | 0.0 | 0 | 0.1 | 37.0 |
| Unexplained weight loss / weight loss | 0.9 | 39.9 | 3.0 | 25.1 |

their GP for seven of the ten symptoms. Indeed the level of contact with a GP for at least one symptom (37%) was high in the Danish study. Another study conducted in one region of Denmark (Funen), of adults aged 20+ years, reported a higher proportion of people experiencing having a *lump in the breast* during the preceding 12 months (3.3%), a similar proportion with *blood in stool* (5.7%) and a lower proportion with *cough longer than six weeks* (6.5%), than in our study. These differences highlight the need for country specific information about the prevalence of symptoms experienced in the community, and associated help-seeking behaviour. Our prevalence and contact with GP findings, obtained from an older population responding to questions focused in symptoms possibly indicative of cancer and which included terms such as persistent, are most pertinent to planners of UK-based cancer awareness campaigns.

A postal questionnaire-based study 3,756 individuals aged 50+ years without cancer and recruited from seven general practices in London, South East and North West England asked about the occurrence of 10 cancer red flag symptoms in the last three months. [15]. Like our study, the prevalence of symptoms was high: 46% of participants reported at least one red flag symptom, with individual symptom prevalence ranging between 2.9% (*unexplained bleeding*) and 16.9% (*cough or hoarseness*). Similar to our study, there was a wide variation in the proportion of symptoms resulting in GP help-seeking: overall 67% of all symptoms were presented, and individually between 53.5% (*cough or hoarseness*) and 72.0% (*unexplained lump*) of symptoms.

The Funen study also looked at factors associated with the reporting of at least one cancer red flag symptom [45]. It found that women, individuals not in the workforce (including because of disability), and those with a cancer diagnosis were more likely to report at least one red flag symptom; with older participants and those living with a partner less likely. Apart from age, these associations are consistent with our findings.

## Implications

Cancer awareness campaigns can increase the proportion of symptomatic individuals seeking help. For example, a national eight week campaign in England in 2012, focused on persistent or prolonged cough as a prompt to seek help to avoid lung cancer [46]. It resulted in a 3% increase in the public's awareness of the potential importance of this symptom (from 12% to

15%), and led to a 67% increase during the campaign in patients of all ages visiting their GP with a cough- equivalent to six extra consultations per practice per week.

Our results highlight the potential implications of such campaigns for healthcare services, particularly general practice. In our study, 13% of respondents had a persistent cough in the previous year, half of whom saw their GP. If those not currently seeing their GP responded positively to a cancer awareness campaign, 6.5% (1 in 15) of all adults in the UK aged 50 + might contact their GP for help. Better understanding of how and why people with particular characteristics respond to symptoms experienced, should enable the tailoring of messages within cancer awareness campaigns for greater effectiveness.

## Conclusion

Symptoms possibly indicative of cancer are common among adults aged 50+ years in the UK, although they are not evenly distributed. Help- seeking responses to different symptoms also vary. Our results suggest important opportunities to provide more nuanced messaging and targeting of symptom-based cancer awareness campaigns.

## Supporting information

**S1 Table. Response rates in Scotland and England.**
(DOCX)

**S2 Table. Proportion and chance of participants in different subgroups having experienced at least one symptom possibly indicative of different cancers in the last year.**
(DOCX)

**S1 File. Questionnaire content.**
(DOCX)

## Acknowledgments

The study was funded by a project grant (C45810/A17927) from Cancer Research UK. We thank the primary care research networks, practices and members of the public who enabled the study to happen.

## Author Contributions

**Conceptualization:** Philip C. Hannaford, Katriina L. Whitaker, Alison M. Elliott.

**Data curation:** Alison M. Elliott.

**Formal analysis:** Philip C. Hannaford.

**Funding acquisition:** Philip C. Hannaford, Peter Murchie, Katriina L. Whitaker, Alison M. Elliott.

**Investigation:** Alison J. Thornton.

**Methodology:** Philip C. Hannaford, Alison J. Thornton, Peter Murchie, Katriina L. Whitaker, Alison M. Elliott.

**Project administration:** Alison M. Elliott.

**Supervision:** Alison M. Elliott.

**Validation:** Alison M. Elliott.

Writing – **original draft:** Philip C. Hannaford.

Writing – **review & editing:** Philip C. Hannaford, Alison J. Thornton, Peter Murchie, Katriina
L. Whitaker, Rosalind Adam, Alison M. Elliott.

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
