## [Decision Letter · Decision Letter 0]

19 Nov 2019

PONE-D-19-28463

Patterns of symptoms possibly indicative of cancer and associated help-seeking behaviour in a large sample of  United Kingdom residents - the USEFUL study

PLOS ONE

Dear Professor Hannaford,

Thank you for submitting your manuscript to PLOS ONE. After careful consideration, we feel that it has merit but does not fully meet PLOS ONE’s publication criteria as it currently stands. Therefore, we invite you to submit a revised version of the manuscript that addresses the points raised during the review process.

Please submit a point-by-point response to each comment raised in the reviews. Please also ensure that your revised manuscript conforms with STROBE reporting criteria for observational studies. 

We would appreciate receiving your revised manuscript by Jan 03 2020 11:59PM. To enhance the reproducibility of your results, we recommend that if applicable you deposit your laboratory protocols in protocols.io, where a protocol can be assigned its own identifier (DOI) such that it can be cited independently in the future. For instructions see: http://journals.plos.org/plosone/s/submission-guidelines#loc-laboratory-protocols

We look forward to receiving your revised manuscript.

Kind regards,

Erin Bowles

Academic Editor

PLOS ONE

Journal Requirements:

1. Please provide additional details regarding participant consent. In the ethics statement in the Methods and online submission information, please ensure that you have specified (1) whether consent was informed and (2) what type you obtained (for instance, written or verbal, and if verbal, how it was documented and witnessed). If your study included minors, state whether you obtained consent from parents or guardians. If the need for consent was waived by the ethics committee, please include this information.

Reviewers' comments:

Reviewer's Responses to Questions

**Comments to the Author**

1. Is the manuscript technically sound, and do the data support the conclusions?

Reviewer #1: Yes

Reviewer #2: Yes

2. Has the statistical analysis been performed appropriately and rigorously? 

Reviewer #1: I Don't Know

Reviewer #2: Yes

3. Have the authors made all data underlying the findings in their manuscript fully available?

Reviewer #1: No

Reviewer #2: Yes

4. Is the manuscript presented in an intelligible fashion and written in standard English?

Reviewer #1: Yes

Reviewer #2: Yes

5. Review Comments to the Author

Reviewer #1: This is a well written manuscript with an interesting research question with many implications for improving cancer awareness research. However, there are some concerns that need to be addressed.

1. There is a mention of focus on red flag symptoms in the introduction but there were not enough details describing what these red flag symptoms were and how they were determined.

2. The study was described as a large community based sample, however more details about the study sampling method should be described to fully understand the sampling population. How did the practices who participated differ from those who did not participate? Moreover, while the reasons for exclusions and number of exclusions were not collected, this is critical for the generalizability of the study and should be addressed further in the discussion.

3. While the questionnaire was a useful instrument for measuring the symptoms in this study, is there any information available about the validity and reliability measures? How well does the questionnaire measure what it is supposed to measure? What is the construct validity of the questionnaire?

4. How were the variables in the models chosen for the analyses? There is no description of how selection of variables were conducted or how they were entered into the models.

5. It would be interesting to understand how the participants over the three mailings might have differed in any of the symptoms. Whether those who needed more reminding differed in any of the characteristics that may be related to the symptoms.

6. It was reported that masking symptoms was critical to minimize any biases in the responses, however some participants contacted the study to clarify the purpose. It would be important to know how many participants contacted the study and if it is of concern.

7. While this research question is interesting, it is still puzzling to understand how some of these symptoms for cancer may be differentiated from a lot of other chronic diseases. Also, it may be important to have information about how many participants actually receive any cancer screening in this population.

Reviewer #2: Thank you for the opportunity to review this interesting paper from a very knowledgeable author group.

The study aims to improve the understanding of the prevalence, patterning and response to symptoms associated with breast, colorectal, lung and upper gastrointestinal cancer in older adults living in the UK general population.

The study adds information to the field, although studies of symptom experiences and healthcare seeking behavior in the population in Western Countries, among others UK and Denmark, have already been published in recent years.

The authors argue that enhanced understanding of the prevalence of symptoms possibly indicative of cancer in different population subgroups, and associated help-seeking behaviour, will help to target cancer awareness campaigns more effectively.

The paper is well written overall and to presents novel data of the UK general population.

The author group describe that their results suggest important opportunities to provide more nuanced messaging and targeting of symptom-based cancer awareness campaigns. After having read the paper, I am though left with the impression that they do not really address this in the discussion. The paper is in that way more descriptive than analytic.

A few methodological issues are raised below:

Related to the questionnaire development: How was ‘persistent’ explained for the respondents? Did the authors make any attempts to qualify how respondents interpreted the questions, and probably also how the interpretations differed among respondents?

In line with this question: How was unexplained interpreted?

I wonder why the author group choose to ask about symptom experiences in the last year. In the literature it is argued that recall bias is a huge problem when asking for more than 6-8 weeks back. Moreover, recall about help-seeking in the past year might be problematic. Some considerations on this issue should be added to the paper

6. PLOS authors have the option to publish the peer review history of their article (what does this mean?). If published, this will include your full peer review and any attached files.

Reviewer #1: No

Reviewer #2: No

---

## [Author Response · Author response to Decision Letter 0]

19 Dec 2019

PONE-D-19-28463

Patterns of symptoms possibly indicative of cancer and associated help-seeking behaviour in a large sample of United Kingdom residents - the USEFUL study

PLOS ONE

Dear Professor Hannaford,

Thank you for submitting your manuscript to PLOS ONE. After careful consideration, we feel that it has merit but does not fully meet PLOS ONE’s publication criteria as it currently stands. Therefore, we invite you to submit a revised version of the manuscript that addresses the points raised during the review process.

Please submit a point-by-point response to each comment raised in the reviews. Please also ensure that your revised manuscript conforms with STROBE reporting criteria for observational studies. 

We would appreciate receiving your revised manuscript by Jan 03 2020 11:59PM. To enhance the reproducibility of your results, we recommend that if applicable you deposit your laboratory protocols in protocols.io, where a protocol can be assigned its own identifier (DOI) such that it can be cited independently in the future. For instructions see: http://journals.plos.org/plosone/s/submission-guidelines#loc-laboratory-protocols

• A rebuttal letter that responds to each point raised by the academic editor and reviewer(s). This letter should be uploaded as separate file and labeled 'Response to Reviewers'.

• A marked-up copy of your manuscript that highlights changes made to the original version. This file should be uploaded as separate file and labeled 'Revised Manuscript with Track Changes'.

• An unmarked version of your revised paper without tracked changes. This file should be uploaded as separate file and labeled 'Manuscript'.

This has been done

We are happy for the peer review history to be released.

 We look forward to receiving your revised manuscript.

Kind regards,

Erin Bowles

Academic Editor

PLOS ONE

Journal Requirements:

This has been done. 

1. Please provide additional details regarding participant consent. In the ethics statement in the Methods and online submission information, please ensure that you have specified (1) whether consent was informed and (2) what type you obtained (for instance, written or verbal, and if verbal, how it was documented and witnessed). If your study included minors, state whether you obtained consent from parents or guardians. If the need for consent was waived by the ethics committee, please include this information.

We have moved the ethical opinion statement to the start of the methods section, renamed it Ethics statement, and added the sentence ‘All participants received written information about the study; participants were deemed to have given consent to participate by returning a completed questionnaire’. 

Reviewers' comments:

Reviewer's Responses to Questions

Comments to the Author

1. Is the manuscript technically sound, and do the data support the conclusions?

Reviewer #1: Yes

Reviewer #2: Yes

2. Has the statistical analysis been performed appropriately and rigorously? 

Reviewer #1: I Don't Know

Reviewer #2: Yes

3. Have the authors made all data underlying the findings in their manuscript fully available?

We are keen to share data wherever possible (indeed we have already done so with a PhD student). In our application for funding we made the following statements:

NAEDI GRANT DATA SHARING PLAN

Cancer Research UK Data Sharing Guidelines will be adhered to at all times. 

Data generated by the proposed grant will be considered for sharing and made as widely accessible as possible whilst safeguarding intellectual property, the privacy of patients and confidential data. 

All data will be stored anonymously and ethically in accordance with the Data Protection Act (1988) and the University of Aberdeen’s code of good practice for maintaining confidentiality of information. 

As our data involves human participants we will ensure that consent is obtained to share information and that the necessary permissions regarding data sharing are in place prior to disclosing any data. 

The research team will consider all requests for data sharing and release data as appropriate. Those requesting data will be asked to provide a brief research proposal on how they wish to use the data and will be asked to sign a data sharing agreement. We will ensure that data is made available in a timely fashion, with the timescale dependent on the stage of study. Data will not be released until the main findings of each phase of the work have been published. 

We will follow the principles of ‘metadata’ when developing our datasets to ensure data released have all the necessary information for users to have a clear understanding of what the data mean so that they can be used appropriately. 

Data will be kept for a minimum of 5 years after the completion of the grant to ensure that the data collected is available for research governance audits and any follow-up research studies. 

We do not expect that the data we collect during this research will have the potential to be exploited commercially, but will monitor and adapt our data sharing plan as the need arises.

Thus, we are encourage people to contact us with a request for access to the data. 

Reviewer #1: No

Reviewer #2: Yes

4. Is the manuscript presented in an intelligible fashion and written in standard English?

Reviewer #1: Yes

Reviewer #2: Yes

5. Review Comments to the Author

Reviewer #1: This is a well written manuscript with an interesting research question with many implications for improving cancer awareness research. However, there are some concerns that need to be addressed.

1. There is a mention of focus on red flag symptoms in the introduction but there were not enough details describing what these red flag symptoms were and how they were determined.

In addition to identifying the red flag symptoms in table 1 we have listed them in the methods section and defined them. We have also indicated that those selected were based on those highlighted in cancer referral guidelines.. 

2. The study was described as a large community based sample, however more details about the study sampling method should be described to fully understand the sampling population. How did the practices who participated differ from those who did not participate? Moreover, while the reasons for exclusions and number of exclusions were not collected, this is critical for the generalizability of the study and should be addressed further in the discussion.

This is an important issue which we have expanded upon in the discussion: ‘We did not have information about how many practices approached by the research networks declined to participate, or about the characteristics of participating practices compared to non-participating practices. In addition, we have no information about the number of, and reasons for, exclusions made by the participating practices when screening the lists of potential invitees. We do not know therefore how generalizable our results are to the wider UK population’. 

3. While the questionnaire was a useful instrument for measuring the symptoms in this study, is there any information available about the validity and reliability measures? How well does the questionnaire measure what it is supposed to measure? What is the construct validity of the questionnaire?

We have added some additional text about this in the opening paragraph of the strengths and limitations section of the discussion. ‘The questionnaire underwent detailed refinement during both the development phase when experts and lay members reviewed early drafts and the pilot phase which resulted in some changes to make the questionnaire easier to follow. A key consideration when discussing early drafts was whether our symptom descriptors accorded with lay understanding. The symptom components of the questionnaire were based on a questionnaire used in a previous study which had been found to collect appropriate data [17, 18]. Questions about the specific concepts of illness perceptions and cancer awareness were taken from validated questionnaires (Brief Illness Perception Questionnaire [29] and Cancer Awareness Measure [16]).

4. How were the variables in the models chosen for the analyses? There is no description of how selection of variables were conducted or how they were entered into the models. I

We have added a clarification to the methods section that the variables included in the model were chosen a priori on the basis of previous research. We have also clarified that all of the variables were entered simultaneously into the model. 

5. It would be interesting to understand how the participants over the three mailings might have differed in any of the symptoms. Whether those who needed more reminding differed in any of the characteristics that may be related to the symptoms.

We agree this is an interesting issue and we had intended looking at it in a subsequent methodological paper. Given the already large amount of information contained in our paper we do not think it is appropriate to include this analysis here.

6. It was reported that masking symptoms was critical to minimize any biases in the responses, however some participants contacted the study to clarify the purpose. It would be important to know how many participants contacted the study and if it is of concern.

We have altered this section slightly to clarify the situation- we do not know exactly how many of the questionnaire recipients thought our questionnaire was about cancer, although we do know that three contacted us specifically to ask about this. The overall number of people who thought this was about cancer is likely to be small. 

7. While this research question is interesting, it is still puzzling to understand how some of these symptoms for cancer may be differentiated from a lot of other chronic diseases. Also, it may be important to have information about how many participants actually receive any cancer screening in this population

We have added some clarification to the discussion paaraghp starting ‘Masking symptoms..’ [original text in blue] … Although some participants reported a history of cancer when completing the questionnaire. We have no information about cancer screening within the study sample. We found that the inclusion of people with a history of cancer did not is unlikely to have materially affected our prevalence results. Some of the reported symptoms are likely to arisen other chronic diseases. Furthermore Nevertheless, most cancer awareness campaigns are targeted at the total population regardless of previous or current medical history, so understanding the prevalence and pattern of cancer-related symptoms, and associated help-seeking behaviour, in the total population is essential…... 

Reviewer #2: Thank you for the opportunity to review this interesting paper from a very knowledgeable author group.

The study aims to improve the understanding of the prevalence, patterning and response to symptoms associated with breast, colorectal, lung and upper gastrointestinal cancer in older adults living in the UK general population.

The study adds information to the field, although studies of symptom experiences and healthcare seeking behavior in the population in Western Countries, among others UK and Denmark, have already been published in recent years.

The authors argue that enhanced understanding of the prevalence of symptoms possibly indicative of cancer in different population subgroups, and associated help-seeking behaviour, will help to target cancer awareness campaigns more effectively.

The paper is well written overall and to presents novel data of the UK general population.

The author group describe that their results suggest important opportunities to provide more nuanced messaging and targeting of symptom-based cancer awareness campaigns. After having read the paper, I am though left with the impression that they do not really address this in the discussion. The paper is in that way more descriptive than analytic.

A few methodological issues are raised below:

Related to the questionnaire development: How was ‘persistent’ explained for the respondents? Did the authors make any attempts to qualify how respondents interpreted the questions, and probably also how the interpretations differed among respondents?

In line with this question: How was unexplained interpreted?

Persistent symptoms: We did not explain or define persistent in the questionnaire. This was intentional.

My PhD revealed considerable variation in what people mean by ‘persistent’. I therefore explored this further in the USEFUL interviews. So we will be able to shed some light on this in the future. We can’t comment on how it was interpreted from the questionnaire, but this is one the strengths of the mixed methods approach, which might be worth highlighting.

Nor can we comment on how participants interpreted ‘unexplained’. Interesting though – perhaps one to explore in future studies? 

We have clarified this issue in the discussion paragraph starting ‘As with all other investigations…. As with all other investigations of symptoms, our study was based on self-reported information. Deliberately, we did not define or explain symptom descriptors such as ‘persistent’ or ‘unexplained’. It is likely that these descriptors have different meanings for different people. In a separate exercise we have conducted qualitative interviews with a number of participants to examine their perceptions of the term ‘persistence’. These findings will be published in a separate paper. Importantly, our study shows how many people perceived themselves to have such symptoms, and how they responded. We have found that participant self-report of help-seeking for symptoms possibly indicative of cancer is reasonably accurate…..

I wonder why the author group choose to ask about symptom experiences in the last year. In the literature it is argued that recall bias is a huge problem when asking for more than 6-8 weeks back. Moreover, recall about help-seeking in the past year might be problematic. Some considerations on this issue should be added to the paper

We have replaced the previous text with the following: We looked at symptom experience over the past month (the main focus) and past year. The patterns of associations were the same regardless of the time frame used. It was useful to include both timeframes to ensure our findings were comparable to previous research in other countries [13]. 

We have also changed the abstract to give one month prevalence figures for the least and most common symptoms, reflecting our focus on one month data.

6. PLOS authors have the option to publish the peer review history of their article (what does this mean?). If published, this will include your full peer review and any attached files.

Do you want your identity to be public for this peer review? For information about this choice, including consent withdrawal, please see our Privacy Policy.

Reviewer #1: No

Reviewer #2: No

---

## [Editor Report · Decision Letter 1]

7 Jan 2020

Patterns of symptoms possibly indicative of cancer and associated help-seeking behaviour in a large sample of  United Kingdom residents - the USEFUL study

PONE-D-19-28463R1

Dear Dr. Hannaford,

We are pleased to inform you that your manuscript has been judged scientifically suitable for publication and will be formally accepted for publication once it complies with all outstanding technical requirements.

With kind regards,

Erin Bowles

Academic Editor

PLOS ONE
---

## [Editor Report · Acceptance letter]

16 Jan 2020

PONE-D-19-28463R1 

Patterns of symptoms possibly indicative of cancer and associated help-seeking behaviour in a large sample of  United Kingdom residents - the USEFUL study 

Dear Dr. Hannaford:

I am pleased to inform you that your manuscript has been deemed suitable for publication in PLOS ONE. Congratulations! Your manuscript is now with our production department. 

With kind regards,

on behalf of

Dr. Erin Bowles 

Academic Editor

PLOS ONE